# STABILIZING OFF-POLICY REINFORCEMENT LEARNING WITH CONSERVATIVE POLICY GRADIENTS

## ABSTRACT

In recent years, advances in deep learning have enabled the application of reinforcement learning algorithms in complex domains. However, they lack the theoretical guarantees which are present in the tabular setting and suffer from many stability and reproducibility problems [10]. In this work, we suggest a simple approach for improving stability in off-policy actor-critic deep reinforcement learning regimes. Experiments on continuous action spaces, in the MuJoCo control suite, show that our proposed method reduces the variance of the process and improves the overall performance across most domains.

## 1    INTRODUCTION

Reinforcement Learning (RL) is a dynamical learning paradigm, in which the algorithm (also known as the 'agent') learns through sequential interaction with the environment. On each round, the agent performs an action, which transitions it into a new state, and is provided with a state-dependent reward. The goal of the agent is to maximize the cumulative reward it observes throughout these interactions with the environment. When considering tabular RL, in which there is a finite number of states and actions, there exist efficient algorithms with theoretical convergence and stability guarantees [29; 13; 14; 34; 5; 4]. However, this is not the case when considering deep RL approaches – when a neural network is used to cope with large state spaces, such as images [19; 11], and/or large action spaces, for example in continuous control [17; 27].

The issue with deep RL is twofold: (i) the optimization process is finite, and as such does not follow many of the requirements for the process to converge, such as decaying learning rates [2], and (ii) the non-linearity of the function approximators (e.g., neural networks) results in a highly non-convex optimization landscape. Due to this, as can be seen in all empirical works [22; 26; 17; 6; 9], the learning process exhibits high variance; during a short time of training, a near-optimal policy may become arbitrarily bad. As a result, in recent years, the stability of RL algorithms has become a major concern of the research community [10].

Instead of tackling the stability problems in the optimization process, many previous works focused on different aspects of the loss function [6; 9; 30]. Then, they usually apply a standard variant of gradient descent algorithm on the modified loss function. While these approaches are capable of finding relatively good policies, they are strictly inferior when compared to their tabular counterparts – the tabular approaches ensure convergence and stability, whereas the practical variants exhibit instability and high variance.

In this work, we suggest a novel approach that improves the stability of actor-critic methods. Specifically, we draw the connection between deep RL approaches and their tabular counterparts and highlight the issues inherent in deep RL algorithms. Through this connection, we propose a solution, in the form of a hierarchical training procedure. As opposed to the tabular case, our approach is not ensured to converge, but rather has stability guarantees, i.e., the stationary distribution of the policies is shown to be stable, meaning that with high probability the performance of the policy improves and does not suffer high degradation. We show, empirically across a range of continuous control tasks in MuJoCo, that our approach indeed results in improved behavior both in terms of stability and overall performance.

## 2    BACKGROUND AND NOTATION

We consider an infinite-horizon discounted Markov Decision Process [23, MDP]. An MDP is defined as the 5-tuple $(\mathcal{S}, \mathcal{A}, P, r, \gamma)$, where $\mathcal{S}$ is the state space, $\mathcal{A}$ the action space,

---

**Algorithm 1** Policy Iteration

---

1: Initialize $\pi_0$
2: $k = 0$
3: **while** not converged **do**
4:     **while** not converged **do**                                                  ▷ Policy Evaluation
5:         **for** $\mathbf{s} \in \mathcal{S}$ **do**
6:             $v^{\pi_k}(\mathbf{s}) = r(\mathbf{s}, \pi_k(\mathbf{s})) + \gamma \sum_{\mathbf{s}' \in \mathcal{S}} P(\mathbf{s}, \pi_k(\mathbf{s}), \mathbf{s}') v^{\pi_k}(\mathbf{s}')$
7:     **for** $\mathbf{s} \in \mathcal{S}$ **do**                                            ▷ Policy Improvement
8:         $\pi_{k+1}(\mathbf{s}) \in \arg\max_{\mathbf{a} \in \mathcal{A}} r(\mathbf{s}, \mathbf{a}) + \gamma \sum_{\mathbf{s}' \in \mathcal{S}} P(\mathbf{s}, \mathbf{a}, \mathbf{s}') v^{\pi_k}(\mathbf{s}')$
9:     $k = k + 1$
10: **return** policy $\pi_k$

---

**Algorithm 2** DDPG, TD3 and our Conservative update

---

1: **Input:** algorithm, learning rate $\alpha$, averaging parameter $\tau$, steps $T$ and exploration $\mathcal{N}$.
2: Initialize policy $\theta$, critics $\phi^1, \phi^2$ and replay buffer $R$
3: Set $\theta_- = \theta, \phi^i_- = \phi^i, t = 0$
4: **while** $t < T$ **do**
5:     Collect data and append to replay buffer $R$
6:     $t = t + 1$
7:     $\{\mathbf{s}_i, \mathbf{a}_i, r_i, \mathbf{s}_{i+1}\} = R.\text{sample}()$                                 ▷ Training phase
8:     **if** DDPG **then**
9:         $\phi = \phi - \alpha \nabla_\phi \| r_i + \gamma Q_{\phi_-}(\mathbf{s}_{i+1}, \pi_{\theta_-}(\mathbf{s}_{i+1})) - Q_\phi(\mathbf{s}_i, \mathbf{a}_i) \|_2^2$
10:     **else**                                                        ▷ TD3
11:         $\phi_1 = \phi_1 - \alpha \nabla_{\phi^1} \| r_i + \gamma \min_i Q_{\phi^i_-}(\mathbf{s}_{i+1}, \pi_{\theta_-}(\mathbf{s}_{i+1})) - Q_{\phi^1}(\mathbf{s}_i, \mathbf{a}_i) \|_2^2$
12:         $\phi_2 = \phi_2 - \alpha \nabla_{\phi^2} \| r_i + \gamma \min_i Q_{\phi^i_-}(\mathbf{s}_{i+1}, \pi_{\theta_-}(\mathbf{s}_{i+1})) - Q_{\phi^2}(\mathbf{s}_i, \mathbf{a}_i) \|_2^2$
13:     $\theta = \theta + \alpha \nabla_\theta Q_{\phi^1}(\mathbf{s}_i, \pi_\theta(\mathbf{s}_i))$
14:     $\phi^i_- = (1 - \tau)\phi^i_- + \tau\phi^i$                             ▷ Update target networks
15:     **if** Conservative Update **then**
16:         **if** $t \% E_{\text{freq}} == 0$ **then**
17:             evaluate $\pi_\theta$ and $\pi_{\theta_-}$ for $E_{\text{num}}$ episodes each
18:             **if** $P(J^{\pi_\theta} > J^{\hat{\pi}_{\theta_-}}) \geq 1 - p$ **then**
19:                 $\theta_- = \theta$
20:     **else**
21:         $\theta_- = (1 - \tau)\theta_- + \tau\theta$

---

*Colored lines represent similarities between the classic and modern approach.*

$P : \mathcal{S} \times \mathcal{S} \times \mathcal{A} \mapsto [0,1]$ is a transition kernel, $r : \mathcal{S} \times A \to [r_{\min}, r_{\max}]$ is a reward function and $\gamma \in (0,1)$ is the discount factor.

The goal of an RL agent is to learn a policy $\pi(\mathbf{s}, \mathbf{a})$ that maps states into a distribution over actions. The quality of a policy is measured by its value, $v^\pi(\mathbf{s}) = \mathbb{E}^\pi[\sum_{t=0}^\infty \gamma^t r(\mathbf{s}_t, \mathbf{a}_t) \mid \mathbf{s}_0 = \mathbf{s}]$, which is the average cumulative reward when starting from state $\mathbf{s}$ and acting according to $\pi$. Another important quantity is the Q-function, $Q^\pi(\mathbf{s}, \mathbf{a}) = \mathbb{E}^\pi[\sum_{t=0}^\infty \gamma^t r(\mathbf{s}_t, \mathbf{a}_t) \mid \mathbf{s}_0 = \mathbf{s}, \mathbf{a}_0 = \mathbf{a}]$, which is strongly connected to the value through the relation $v^\pi(\mathbf{s}) = \int_{\mathbf{a}} Q^\pi(\mathbf{s}, \mathbf{a})\pi(\mathbf{s}, \mathbf{a})d\mathbf{a}$. We denote an optimal policy that maximizes the value simultaneously across all of the states by $\pi^* \in \arg\max_\pi v^\pi$ and the optimal value function by $v^* = v^{\pi^*}$. When the Q-function and the policy $\pi$ are represented by a parametric function, we denote them by $Q_\phi$ and $\pi_\theta$, where $\phi$ and $\theta$ are the parameters of the function (e.g. the parameters of a deep neural network).

While our goal is to maximize $v^\pi$, in practice we measure an empirical estimation of $\mathbb{E}_{\mathbf{s} \sim \rho} v^\pi(\mathbf{s})$, where $\rho$ is the initial state distribution. To accommodate for this fact, we define the cumulative reward of a sampled trajectory by $J^\pi$. This is a random variable, as randomness may occur due to (i) the initial state distribution, (ii) the transition kernel and reward function and (iii) the policy.

## 2.1 CLASSIC REINFORCEMENT LEARNING ALGORITHMS

Policy Iteration [12, PI] is an algorithm for solving tabular MDPs with convergence guarantees. PI (Algorithm 1) iterates between two steps: (i) *policy evaluation* in which the current policy $\pi_k$ is evaluated, producing the value $v^{\pi_k}$ and (ii) *policy improvement* in which the policy $\pi_{k+1}$ is updated greedily w.r.t. $v^{\pi_k}$. This iterative scheme is proven to converge to the optimal policy.

PI schemes have also been extended to the approximate case. Scherrer [25] provides an overview of such approaches, which are called Approximate Policy Iteration (API). In API we consider the scenario in which $||\hat{v}^{\pi_k} - v^{\pi_k}||_\infty \le \epsilon$. This approximation error can be caused due to two factors, the first being the inability to estimate the *value* and the second being the inability to find the *greedy policy*, i.e., the $\arg\max$ at each state. For instance, such errors may occur due to the functional class being used, e.g., neural networks. Most importantly, the sub-optimality of these approaches is bounded by $||v^{\hat{\pi}_\infty} - v^*||_\infty \le \frac{\epsilon}{(1-\gamma)^2}$ [25], i.e., the sub-optimality of the final policy is proportional to the approximation error $\epsilon$.

## 2.2 OFF-POLICY DEEP REINFORCEMENT LEARNING FOR CONTINUOUS CONTROL

DDPG [17] and its latest improvement TD3 [6] use neural networks as function approximators in order to find the optimal policy. Both approaches utilize a target network, which is crucial for the success of the methods. While in their work, this was merely considered as a design parameter, we argue that the importance of these target networks lies in the connection to Policy Iteration.

**Deep Deterministic Policy Gradients** [17, DDPG] (Algorithm 2) aims to directly learn a deterministic policy in a continuous action space. DDPG has both policy evaluation and policy improvement steps. The $Q$-function, also known as the critic, is trained to estimate the utility of the *target policy* $\pi_{\theta_-}$, whereas the online policy (the actor) is trained to find the 1-step greedy update w.r.t. this estimation. We argue that DDPG and PI are connected through the target network update procedure. In PI, the previous policy $\pi_k$ is evaluated by $v^{\pi_k}$, followed by an improvement step in which $\pi_{k+1}$ becomes the one-step greedy policy w.r.t. this estimate. DDPG follows a similar scheme, in which the critic $Q_\phi$ estimates the utility of the policy $\pi_{\theta_-}$, followed by the online policy $\pi_\theta$ which is updated w.r.t. this estimate. In this work, we compare to TD3 [6], an improved version of DDPG.

**Twin Delayed DDPG** [6, TD3] improves the DDPG algorithm by tackling the problem of overestimation. Similarly to how Double-DQN [32] overcame the overestimation problem by introducing an additional Q function, in TD3 they introduce an additional critic. They show, both theoretically and empirically, that by bootstrapping the minimal value between both estimators $\min_i Q_i(\mathbf{s}, \pi(\mathbf{s}))$ they are able to overcome the problem of overestimation.

## 3 INSTABILITY IN POLICY-BASED DEEP REINFORCEMENT LEARNING

In the previous section, we provided an overview of PI, which has convergence guarantees, and two off-policy DRL variants. However, since in practical applications the policy is represented using a non-linear neural network, a small change in the parameters $\theta$ may result in a very large change in the output. We illustrate this issue in Figure 1, where a state of the art off-policy approach learns to control the Hopper and InvertedPendulum robots.

In these figures, we show the individual learning curves over several seeds, without performing temporal smoothing[1], whereas the bold line represents the mean performance across runs. At each evaluation period, we evaluate the performance of the agent across 100 episodes. These results paint a clear picture – while the agent is capable of learning a policy which attains a score of approximately 3500 in Hopper and the optimal score of 1000 in InvertedPendulum, it is *highly unstable*, i.e., a policy which was near-optimal at time $t$ might become arbitrarily bad at time $t + 1$.

---

[1]Temporal smoothing is a common practice in empirical studies, in which each data-point in the graph represents a moving average over the past N evaluations. While this makes the graphs easier to read, it hides a lot of important information such as the variance of each seed during training.

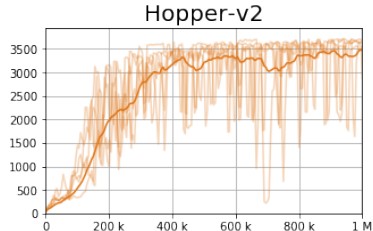 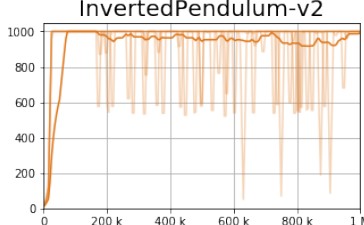

Figure 1: Performance plots of the TD3 [6] algorithm.

## 4  CONSERVATIVE POLICY GRADIENTS

While there exists a connection between the tabular approach and that which is used with non-linear function approximators, due to the non-convex optimization landscape, a small gradient step can lead to an arbitrarily bad solution. As such, current policy-based methods with non-linear function approximation (neural networks) behave poorly when compared to their tabular counterparts.

We propose a simple, conservative, yet theoretically grounded approach for tackling these issues. Taking inspiration from the classic approaches, we observe that the gradient-based algorithms lack the improvement guarantees. To overcome this issue, we propose a hierarchical learning procedure: while the online network is trained as before (as seen in Algorithm 2), the target network is updated periodically if, with high probability, the online policy is better.

In simple terms, *our proposal is as follows:* every $T$ time-steps, evaluate both the target network[2] and the online network. If with confidence of $1 - \delta$ the online network is better than the target, perform the switch $\theta_- = \theta$. In addition, between updates, the target network is kept fixed, as opposed to the standard approach [17; 6] which performs Polyak averaging [21]. We provide additional discussion in Appendix D.

The following proposition motivates this approach. We discretize the range $[v_{\min}, v_{\max}]$ into bins of size $\Delta$ and formulate the learning process as a random-walk, in which with probability $1 - \delta$, the policy improves (1 step to the right) and decreases otherwise. The random-walk is bounded at both ends by $v_{\min}$ and $v_{\max}$, such that the value is projected back into this range.

**Proposition 1.** *Let $v_t^\pi$ be a random walk admitting values in the set $V = \{v_{\min}, v_{\min} + \Delta, \ldots, v_{\max} - \Delta, v_{\max}\}$. Also, assume that w.p. $\delta$: $v_{t+1}^\pi = \min\{v_t^\pi + \Delta, v_{\max}\}$ and $v_{t+1}^\pi = \max\{v_t^\pi - \Delta, v_{\min}\}$ otherwise. Then, the stationary distribution $v$ of $v_t^\pi$ equals $\mathbb{P}(v = x) = r^x \frac{1-r}{1 - r^{(v_{max} - v_{min})/\Delta + 1}}$ for any $x \in V$, where $r = \frac{\delta}{1-\delta}$.*

*Proof Sketch.* We can write the recurrence relations by:

$$\mathbb{P}(v = v_{\min}) = (1 - \delta)\mathbb{P}(v = v_{\min}) + \delta\mathbb{P}(v = v_{\min} + \Delta)$$
$$\mathbb{P}(v = x) = (1 - \delta)\mathbb{P}(v = x - \Delta) + \delta\mathbb{P}(v = x + \Delta)$$
$$\mathbb{P}(v = v_{\max}) = (1 - \delta)\mathbb{P}(v = v_{\max} - \Delta) + \delta\mathbb{P}(v = v_{\max})$$

the stationary distribution follows. □

An illustration of this process is provided in Appendix A. When the probability of improvement is 50%, i.e., we are unable to determine which policy is better, the stationary distribution over the value may be arbitrarily bad. However, as the confidence in the improvement of the policy increases, the distribution shifts towards a delta function at the optimal value function.

**Remark 1.** *In practice, the optimal attainable policy is a function of the initial starting parameters. Due to the non-convexity of the optimization landscape, not all initializations are ensured to be capable of attaining an optimal policy.*

---

[2]The target network needs to be re-evaluated in order to ensure an unbiased estimate of its performance.

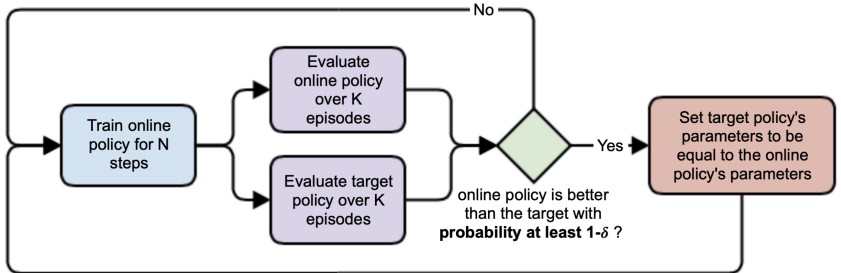

Figure 2: Conservative Policy Gradients.

## 5 EXPERIMENTS

Implementing our approach in an off-policy actor-critic scheme is straightforward. While current approaches [17; 6] hold two policies, an online and a target policy, such that the target policy is updated each step using a Polyak averaging procedure [21] – our approach, illustrated in Fig. 2, requires the target policy to be updated only when the online policy is superior, with high probability.

Since in our framework, the online policy is aimed at finding the 1-step greedy policy, and due to the continuous nature of the action space, we add a 'trust region' element to the loss $\lambda||\pi_\theta(\mathbf{s}) - \pi_{\theta_-}(\mathbf{s})||_2^2$ such that the online policy is kept close to the target. In addition, we found that collecting data interchangeably from the online and the target network leads to a further improvement in stability.

For all the experiments, we plot 5 different random seeds for each configuration, discarding failure seeds[3]. At each evaluation step, each seed is evaluated for 100 episodes, and the average performance is presented. We plot both the individual seeds (light curves) and the average results across seeds (dark curves). As opposed to previous works, we do not smooth the graphs using a moving average procedure, which enables a better analysis of the experiments' variance and stability.

### 5.1 EVALUATING THE POLICIES

Our approach requires determining whether or not an online policy is better, with probability $1 - \delta$. Assuming $J^\pi$, the performance of a sampled trajectory for any deterministic policy $\pi$, is normally distributed, we may consider the $t$-test for evaluating improvement. We formulate the null hypothesis as $\mu_{\pi_\theta} = \mathbb{E}_\rho J^{\pi_\theta} \leq \mathbb{E}_\rho J^{\pi_{\theta_-}} = \mu_{\pi_{\theta_-}}$. In other words, we hypothesize that the online policy has not improved over the target policy, thus we use the Welch's one-tailed t-test [33]. The $t$-statistic is then defined as $t = \left( \hat{\mu}_{\pi_\theta} - \hat{\mu}_{\pi_{\theta_-}} \right) / \left( \sqrt{\hat{\sigma}^2_{\mu_\theta}/n_{\pi_\theta} + \hat{\sigma}^2_{\mu_{\theta_-}}/n_{\pi_{\theta_-}}} \right)$, where $\hat{\mu}_\pi$ is the empirical estimate of $\mathbb{E}_\rho J^\pi$, $\hat{\sigma}_\pi$ is its standard deviation and $n_\pi$ the number of sampled trajectories. The $t$-statistic enables us to find the $p$-value, which represents the confidence in the null hypothesis. Under Gaussian assumption, the $p$-value can be directly calculated by $p = 1 - \Phi(t)$, where $\Phi(x)$ is the cumulative distribution function of a standard normal random variable. Hence, when $p < 1 - \delta$ we have a high enough confidence that the null hypothesis is incorrect - thus the new policy is better, in which case we update the target policy.

### 5.2 PARAMETER SELECTION

For this approach, there are three parameters which require tuning. (i) how often we evaluate the online policy, (ii) how many episodes are used for evaluation and (iii) the minimal confidence required for policy improvement. Notice that there is a connection between the confidence and the number of evaluation episodes; as the number of evaluations grows, the confidence itself increases. Hence, when there is a relatively high variance in the evaluations, a low number of samples will result in a low confidence, even when the empirical mean is higher. On the other hand, the evaluation frequency controls how fast the process converges. We evaluate the effect of these parameters and provide comprehensive results in Appendix B.

---

[3]We observed that some seeds in deep RL result in utter failure, for instance, a score of 40 in Hopper. This issue is relatively rare, and also happens in the baselines, but hinders the proper evaluation of the policy.

**Confidence Value** $\delta$**:** The confidence level is $1 - \delta$, i.e., $\mathbb{P}(\mathbb{E}_\rho J^{\pi_\theta} > \mathbb{E}_\rho J^{\pi_{\theta^-}}) \geq 1 - \delta$. While a natural approach would be to attempt a high level of confidence, our empirical results show otherwise. Demanding a confidence which is too high, makes it much harder to find an improving policy, resulting in a stable yet strictly sub-optimal policy. On the other hand, when very low, the approach becomes unstable. We found that $\delta \in [0.1, 0.2]$ work well in practice.

**Evaluation Episodes** $K$**:** The number of evaluations is correlated to the confidence. A higher number of evaluations increases the confidence in the empirical mean, enabling the algorithm to determine that a policy is better even when the difference is small. However, this comes at a cost – evaluating the policy takes time and a multitude of evaluations may drastically lengthen training.

**Evaluation Frequency** $N$**:** When comparing evaluation frequency, we see that when the evaluation is performed every $10,000$ steps, it requires a high probability of improvement to successfully converge. However, when the evaluation is performed more often, the process behaves well even for a lower improvement probability (fewer evaluation episodes).

Based on these insights and the results in Appendix B, we opted to run with a minimal confidence for swapping of $90\%$, evaluation every $1,000$ steps over 10 episodes (sampled trajectories).

## 5.3 RESULTS

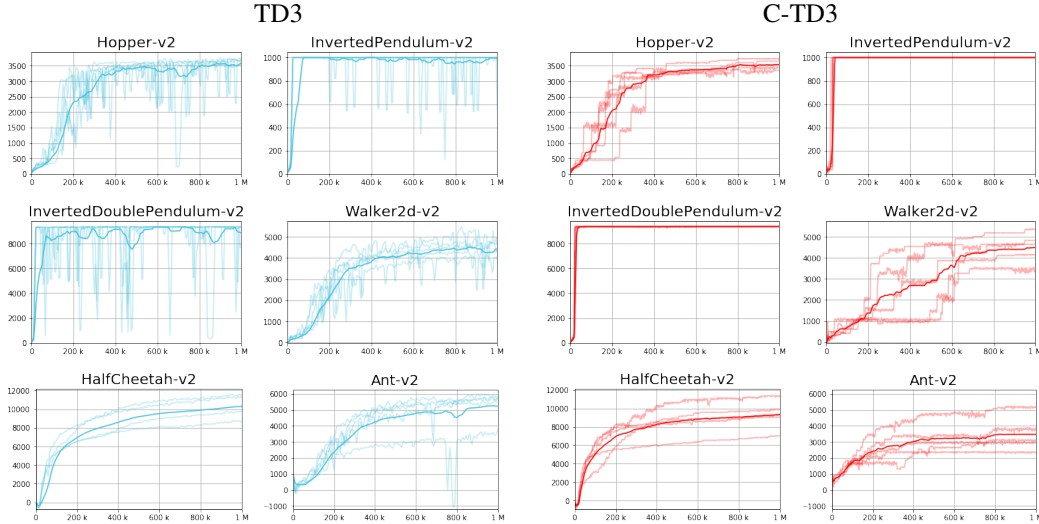

Figure 3: Evaluation without (TD3) and with the Conservative (C-TD3) update rule. X-axis denotes the number of gradient steps taken.

We compared our approach to the baseline – TD3 without our policy selection scheme. The results are presented in Figure 3. These results show a high resemblance to the theoretical model, showing that, across all domains, the conservative approach (Conservative-TD3) reduces the variance of individual seeds, also seen in Table 1, and results in increased stability across most domains.

InvertedPendulum and InvertedDoublePendulum serve as perfect examples. While TD3 is capable of finding an optimal policy - e.g., the maximal score - subsequent policies may be arbitrarily bad. By adding the hierarchical selection scheme, our approach is capable of reducing this behavior. That is, once an optimal policy is found, it is not likely to be replaced with an inferior one.

An interesting result can be seen in the Walker2d domain. While some seeds attain near-optimal performance, others converge to a sub-optimal policy. As our approach searches for a high-confidence improvement, it may sometimes get stuck. However, the inner-process variance, i.e., the variance of individual seeds during training, is much lower across all domains.

Finally, we observe sub-optimal behavior in the Ant domain, both in terms of individual seeds and average performance across seeds. The parameters (confidence, number of evaluation episodes and

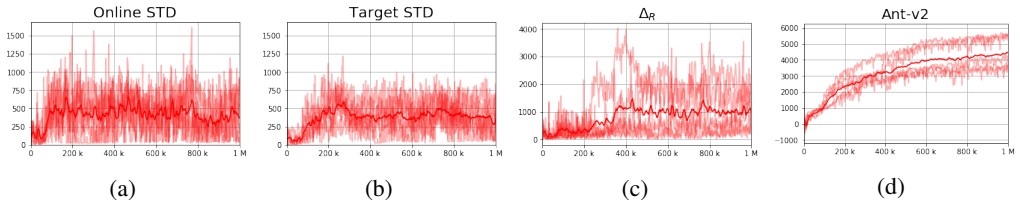

Figure 4: Failure analysis of the Ant-v2 domain.

evaluation frequency) were selected based on the Hopper domain, however it is important to note that these parameters are not optimal for each domain.

Figure 4 presents the standard deviation (STD) of both the online (Figure 4a) and target (Figure 4b) networks, and the difference between the empirical means $|\hat{\mu}_\pi - \hat{\mu}_{\pi_-}|$ (Figure 4c). This clearly shows that in Ant, there is a very large variance, compared the difference between the means, which results in a low confidence. To overcome this issue, we provide an additional experiment, for the Ant domain based on these insights (Figure 4d), which considered a minimal confidence of 20% (90% in Figure 3). These results show that while the parameters are transferable across domains, the optimal parameters found on one domain are not necessarily optimal for another.

### 5.3.1 STABILITY ANALYSIS

|        | Hopper | InvertedPendulum | InvertedDoublePendulum | Walker2d | HalfCheetah | Ant |
|--------|--------|------------------|------------------------|----------|-------------|-----|
| C-TD3  | 36     | 0                | 6                      | 99       | 81          | 57  |
| TD3    | 799    | 139              | 2409                   | 468      | 350         | 902 |

Table 1: Process STD calculated over the last 100 training episodes.

To analyze the stability of each algorithm (TD3 vs C-TD3), we consider the variance of the policy during the last 100 episodes of training. In Table 1 we present the maximum of the standard deviation in different seeds, where per-seed STD is calculated over the 100 last episodes of training.

Our empirical results match the theoretical behavior. Indeed, the inner-seed variance is dramatically reduced when using the conservative update rule, as opposed to the non-conservative TD3.

### 5.4 POTENTIAL EXTENSIONS

In this work, we presented an approach for providing probabilistic improvement guarantees. The performance of a trajectory is a random variable, as the initial state, the transition kernel and sometimes the policy – may all be stochastic. To determine which policy $\pi_-$ or $\pi$ (the target or online) is better, we performed $N$ evaluations, assumed that the samples are normally distributed and performed a $t$-test. Below we present several intuitive extensions to our work:

**Unknown distribution:** While the Gaussian assumption worked well in practice, it may not always be correct. In such a scenario, we see two possible routes: (i) known range, in which the minimal and maximal possible values are known, where one may use the Empirical Bernstein [18], or (ii) when the range is unknown, one may consider a bootstrap-based approach for estimating the confidence intervals around the mean [1].

**Upper/Lower Confidence Bounds:** Another approach is to consider improvement by a minimal margin. For instance, we may set the update rule to $\mathbb{P}(v^\pi - \epsilon \geq v^{\pi_-}) \geq 1 - \delta$, so that the statistical test ensures that with high confidence, the online policy has improved by at least $\epsilon$ over the target.

**Adaptive Sampling:** In our approach, we considered a rather small number of evaluations, and thus resorted to an equal number of evaluations of each policy – up to 100 evaluations for each policy. However, when provided with a larger budget, it is of great interest to combine adaptive sampling procedures. Gabillon et al. [7] propose such an approach, which takes into account the empirical mean and variance of each policy evaluation in order to determine which policy to evaluate. At the end of this process, the policy with the highest empirical mean is returned.

In our initial experiments, we tested the UCB/LCB approach. However, we found that $\epsilon$ is domain-specific. While in some domains the attained rewards are relatively high, and the policies improve greatly between iterations, in others the improvement is rather small. This led to an early convergence to sub-optimal policies. Nevertheless, this approach may prove beneficial in certain settings and should be considered. Additionally, we tested policy swap based on the empirical estimate of the mean, without ensuring a proper confidence. While this approach worked relatively well in practice, as it lacks proper confidence bounds, we decided to focus on the $t$-test based approach.

## 6    RELATED WORK

**Trust Region Optimization:**    Recent works in on-policy RL have also taken inspiration from Conservative Policy Iteration [15, CPI]. Schulman et al. [26] formulated learning through a trust region optimization scheme, which was later optimized in Schulman et al. [27]. These approaches bound the KL-divergence between the policy before and after the update. While TRPO greatly improved the stability compared to previous policy gradient approaches, such as A2C [20], it lacks the theoretical guarantees.

**Safe Policy Improvement:**    Given a baseline policy, safe policy improvement (SPI) algorithms aim to find a new policy that is better than the baseline with high probability [31; 8; 16]. Although this may seem similar to our update stage, there are two major differences: (i) most SPI algorithms directly calculate the improving policy, which is much less efficient than continuously updating the policy and performing the switch when improvement is achieved, and (ii) due to safety constraints, SPI algorithms usually perform off-policy evaluation for the suggested policy, which is still an open problem in high dimensional problems. In contrast, we evaluate it directly and thus enjoy a better sample complexity.

**Evolution Strategies:**    ES [24; 28; 3] is a gradient-free approach for solving reinforcement learning tasks. At each iteration, given the previous policy $\pi_{\theta_i}$, a set of augmented policies is created $\{\pi_{\theta_i + \xi_i}\}_i$ and evaluated, where $\xi_i$ is some random noise. The ES update rule can be seen as a form of numerical gradient estimation (i.e., finite differences) [35]. There are some similarities between our approach and ES. While ES is a gradient-free method, it uses empirical evaluations of the policies to determine the parameter update direction. Our approach uses these evaluations in order to determine whether or not the policy has improved, and in contrast to ES, will only change the policy if there is a high enough confidence for improvement.

## 7    CONCLUSIONS

In this work, we tackled the stability problems of policy-based reinforcement learning approaches. Even though the empirical gradient (SGD) is a noisy estimate of the ideal descent direction, and due to the structure of the optimization landscape and the large step sizes used in practice, the policy's performance varies greatly during training.

Drawing from the connection between actor-critic approaches and approximate policy iteration schemes, we introduced a hierarchical training scheme, which we analyzed by comparing this approach to a bounded random-walk. While this scheme is not an exact representation of the optimization process, it provides an intuition on the behavior of such a probabilistic improvement scheme, i.e., as the probability of improvement increases, the stationary distribution over the performance of the policy shifts towards the optimal solution.

Finally, we proposed the Conservative-TD3, which performs a periodic evaluation of both the online and target policies such that it updates the target network only when there is a higher enough confidence that the online network has surpassed its performance. We validate this on several continuous control tasks in MuJoCo; empirical evidence shows that indeed this approach behaves similarly to the theoretical analysis. When compared to the baseline, we observe a dramatic reduction in variance, increased stability and an overall improvement in the performance itself.

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

# A  RANDOM WALK

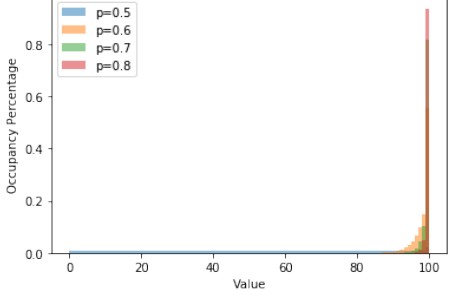

Figure 5: Stationary distribution of a bounded random walk process. As the probability of improving (moving to the right) increases, the distribution shifts towards a delta function around the optimal value.

We expect our approach to approximately behave as a random walk on a bounded region. As the probability of taking a step towards the right increases, the stationary distribution shifts towards the optimal value. On the other hand, as the uncertainty grows, i.e., the probability of improvement becomes closer to $\frac{1}{2}$, the stationary distribution is uniform.

## B    PARAMETER ABLATION TESTS

In this section, we present the simulation results for all possible parameter combinations, in the Hopper domain. Specifically, we tested minimal required improvement probability of $[0.5, 0.8, 0.9, 0.95]$, number of evaluations per evaluation steps of $[5, 10, 100]$ and evaluation frequency of $[1000, 10000]$.

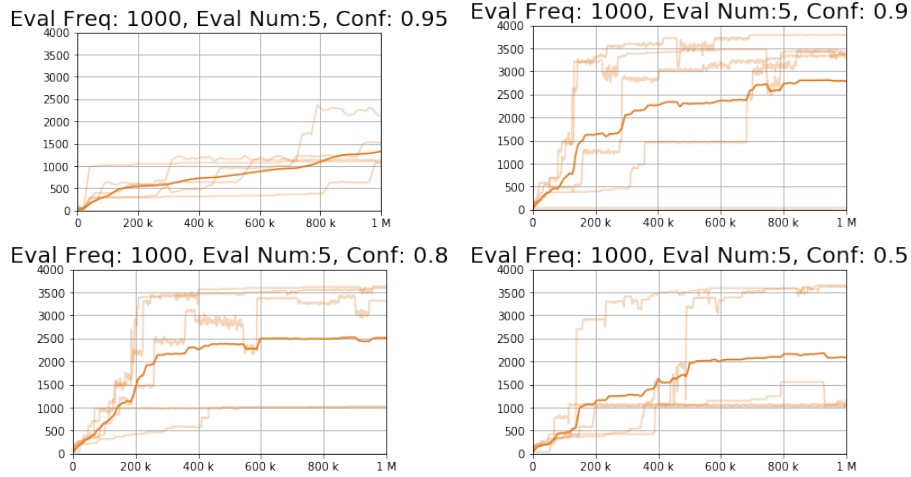

Figure 6: C-TD3, parameter ablation. Comparison is performed every 1,000 steps over 5 episodes with swap confidence of $1 - \delta = 95, 90, 80$ and $50$.

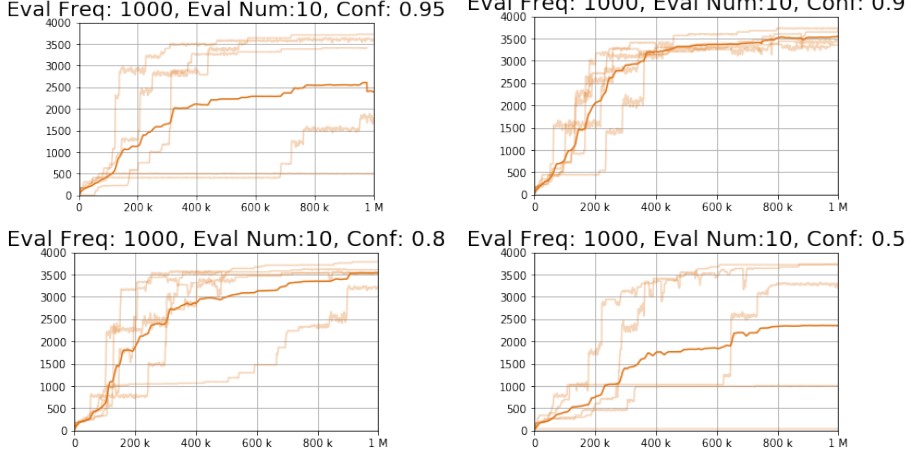

Figure 7: C-TD3, parameter ablation. Comparison is performed every 1,000 steps over 10 episodes with swap confidence of $1 - \delta = 95, 90, 80$ and $50$.

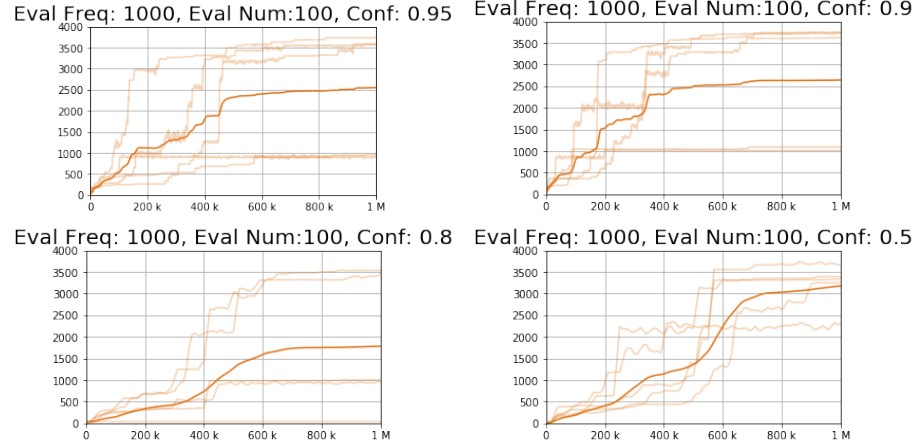

Figure 8: C-TD3, parameter ablation. Comparison is performed every 1,000 steps over 100 episodes with swap confidence of $1 - \delta = 95, 90, 80$ and $50$.

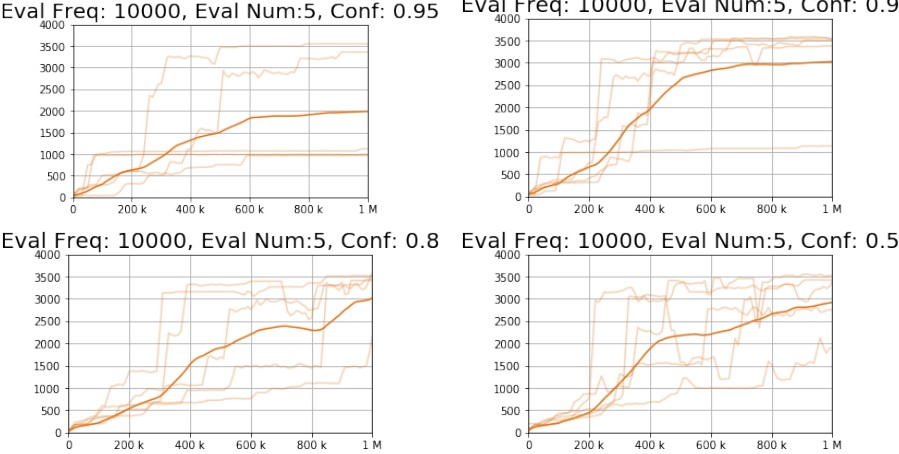

Figure 9: C-TD3, parameter ablation. Comparison is performed every 10,000 steps over 5 episodes with swap confidence of $1 - \delta = 95, 90, 80$ and $50$.

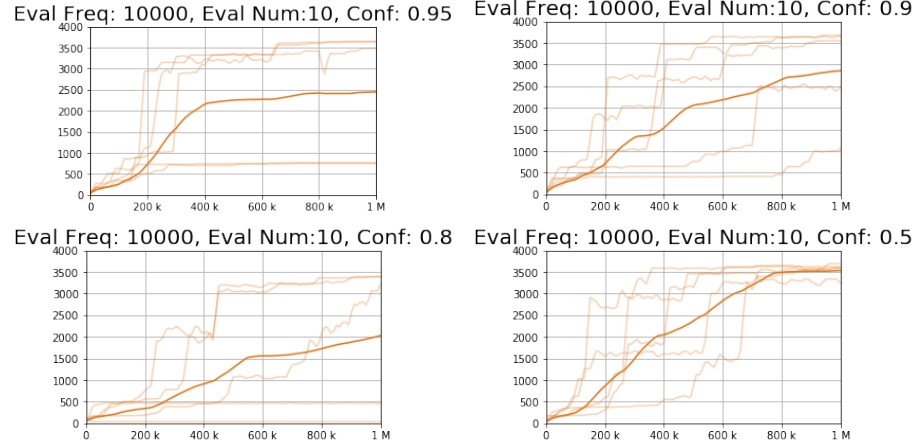

Figure 10: C-TD3, parameter ablation. Comparison is performed every 10,000 steps over 10 episodes with swap confidence of $1 - \delta = 95, 90, 80$ and $50$.

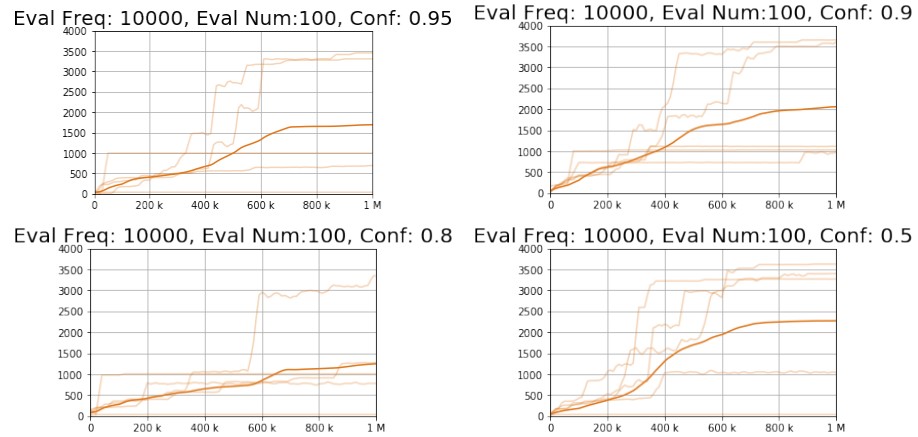

Figure 11: C-TD3, parameter ablation. Comparison is performed every 10,000 steps over 100 episodes with swap confidence of $1 - \delta = 95, 90, 80$ and $50$.

The graphs above show the effect of tuning the evaluation frequency, the number of evaluations and the minimal required confidence. As the evaluation becomes more frequent, the agent learns faster. As our approach requires the critic to evaluate the target policy $\pi_-$, the online policy $\pi$ aims to find the 1-step greedy policy. This may happen rather quickly, thus a more frequent evaluation scheme will improve faster. On the other hand, the number of evaluation episodes is correlated to the confidence in improvement, i.e., the confidence we have that $\pi$ is indeed 'better' than $\pi_-$. We see that as the number of evaluations rises, the agent is capable of finding good solutions even when a higher confidence is required. However, a lower confidence results in a noisier training regime and a small number of evaluations leads to a low confidence – hence high confidence requirements with a low number of evaluations often leads to the inability to converge to a good solution.

## C  ONLINE VS. TARGET NETWORKS

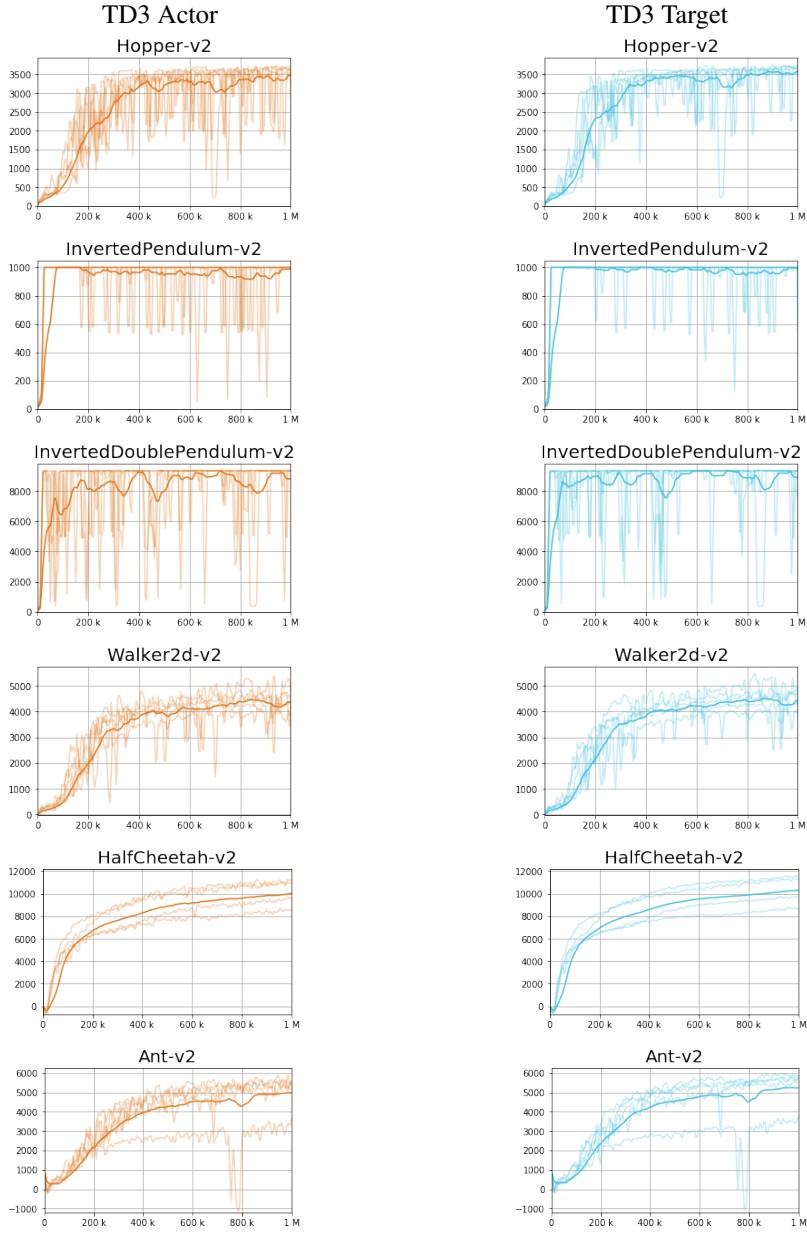

Figure 12: On the left is the evaluation of the online network, whereas on the right we present the performance curves of the target network. It is clear across all domains that the target network exhibits a dramatic reduction in variance.

The common approach is to present the performance of the online network. In these graphs, we show that, as opposed to the common belief, the target network suffers less from instability and offers improved performance. These results are intuitive once the connection to policy iteration schemes is made.

| Parameter | Value | Comment |
|:---:|:---:|:---:|
| $\lambda$ | 0.1 | The trust-region weight. |
| learn_start | 1,000 | Number of steps during which actions are sampled uniformly. |
| swap_start | 10,000 | Initial steps during which training is performed using Polyak Averaging. |
| max_timesteps | 1e6 | Total number of steps during training. |
| expl_noise | 0.1 | Exploration noise variance. |
| policy_noise | 0.2 | Smoothing parameter for the critic training. |
| noise_clip | 0.5 | Clip the noise to [-0.5, 0.5]. |
| policy_freq | 2 | Policy is trained once every two critic training phases. |
| reset_policy_noise | 0.01 | Noise applied to policy when stuck for too long. |

Table 2: Hyper Parameters.

## D EMPIRICAL DETAILS AND DISCUSSION

We follow the same hyper-parameter scheme as is used for the TD3 algorithm [6]. The first $10,000$ steps are played using randomly sampled actions, during which the target policy is updated in a Polyak averaging technique. After which we initialize the conservative periodic policy updates.

In addition, we noticed that while the target policy has a high probability stability assurance, this does not hold for the online policy. As such, we found that often the online policy converges to a strictly sub-optimal solution. We overcome this issue using two heuristics, (i) using a trust-region update rule in which we penalize (an additional loss term) the distance between the action predicted by the online and the target policies. As the critic estimates the performance of the target policy, the estimation error grows as the online policy diverges from the target. Finally, (ii) if the policy has not been updated for over $20,000$ steps, we assume that it has converged to a sub-optimal region of the loss landscape. At such an event, we set the new online parameters as $\theta = \theta + u$ where $u \sim \mathcal{N}(0, 0.1)$ – this is akin to an ES search procedure in which noise is added to the parameters in order to overcome local extrema. This approach helps, but does not ensure, preventing the approach from "flatlining" in which the online policy is no longer updated.

While the online policy benefits from the periodic updating procedure, we observed that when the critic is also updated periodically, the performance becomes strictly sub-optimal. We believe that this is since this introduces an additional time-scale, where three timescale algorithms are known to be tricky to tune and properly operate.

Finally, we observed that in certain initializations both TD3 and C-TD3 were incapable of learning. This resulted in a flat-line throughout the entire training regime, e.g., results such as a score of 40 in Hopper. Although rare, this is indeed an important problem in DRL, which needs to be addressed properly, and is not apparent in supervised learning regimes. However, as this is not the focus of this paper, we discard such seeds to enable the focus on the proper training regimes.

