# OpenReview forum: "Stabilizing Off-Policy Reinforcement Learning with Conservative Policy Gradients"
_ICLR.cc/2020/Conference — Reject_

### Official Review · AnonReviewer2 · 2019-10-22
**Official Blind Review #2**

**Rating:** 3

**Review:**

This paper proposed to use target network policy as a conservative policy for performance evaluation. Instead of performing Polyak averaging on the target network, the authors proposed to utilize statistical hypothesis testing to check whether the performance of the online policy is better than the target network policy and update the target network policy according to the results of the hypothesis testing.

The paper is clear written and easy to follow the core idea. As for the experiments, the authors evaluate the proposed method on a variant of TD3 (Conservative-TD3) and the experimental results indicate the proposed method indeed reduces the variance of the expected return. Ablation studies has been provided in the appendix to show the effectiveness of the proposed method.

Besides the promising results, I believe there are several concerns that should be clarified before we can conclude that the proposed method can improve the stability.

- Stability Measurement: While the experimental results show that the proposed method can reduce the variance of expected return, it is not a direct measurement of the stability or the robustness of the learned policy. It is better to show whether the proposed method can satisfy stability or robustness definition of RL algorithms. (For example, whether the proposed method can improve the robustness: Given a policy, by adding a \epsilon perturbation to the input, the output is still \epsilon-robustness) Otherwise, the author should add some discussion to clarify the difference.

- Conservative Updates on Q functions. In Actor critic frameworks such as DDPG or TD3, the performance of the actor is usually determined by the critic. The proposed method is more likely to “select” stable actors rather than directly improving the stable of the training process or the stability of the policy. I wonder whether it is possible to improve the stability of Q updates such that the “selection” process of policy can be easier, which may accelerate the current training process (or making the hypothesis easier to satisfy).

- More related work should be compared. The authors only compare the original version of TD3 and the modified proposed method. Other recent proposed methods to improve stability should be compared in the experiments, such as Constrained Policy Optimization (Achiam et. al 2017), Lypunov-based Safe Policy methods (such as Chow et. al 2019).

- Noisy Environment. The authors demonstrate the stability of the proposed method in the ordinary mujoco benchmarks. How does the proposed method perform in the noisy MDP settings? Since the original Mujoco implementation is deterministic, the experiments that the author conducted are not enough to show the proposed method can generalize to more realistic settings such as noisy MDPs. It would be more convincing if the method can still perform well in such settings to support the claim.

Overall I think the authors proposed a simple but yet effective method to improve the stability of policy, while the current paper requires more comparison with other methods and more challenging settings to show the effectiveness of the proposed method.

--------------------------------
I will update my score if the author clarify above questions.


-------------------------------
The author clarified one of my main concern, but the other reviewers point out that the comparison is not fair (using only 5 seeds and discarding the failure seeds).

Related Papers:
Achiam, Joshua, et al. "Constrained policy optimization." Proceedings of the 34th International Conference on Machine Learning-Volume 70. JMLR. org, 2017.

Chow, Yinlam, et al. "Lyapunov-based Safe Policy Optimization for Continuous Control." arXiv preprint arXiv:1901.10031 (2019).

**Experience Assessment:**

I have read many papers in this area.

**Review Assessment: Checking Correctness Of Derivations And Theory:**

I assessed the sensibility of the derivations and theory.

**Review Assessment: Checking Correctness Of Experiments:**

I assessed the sensibility of the experiments.

**Review Assessment: Thoroughness In Paper Reading:**

I read the paper at least twice and used my best judgement in assessing the paper.

---

> ### Author Response · Authors · 2019-11-08
> **Official Response**
>
> We would like to thank the reviewer for the thoughtful review. We will address the various concerns below:
>
> - Stability: The notion of stability in this work is a different notion than robustness. While robustness considers the ability to cope with slight perturbations (e.g., uncertainty in the task), we measure the stability of the learning process. For instance, in Table 2 we analyze the variance of the final 100 evaluations of each environment and show that our conservative update rule stabilizes the performance.
>
> - Conservative Updates: Unfortunately we did not understand the comment. Could you please elaborate so we could discuss this further?
>
> - Related work: Safety and stability have certain common aspects but they are fundamentally different. Chow et. al. and Achiam et. al. consider the scenario in which there are behavioral constraints on the policy - such as regions which the agent should not enter. Our approach does not limit the policy in any way, but rather aims to ensure that the performance is monotonically increasing.
>
> - Noisy Environment: This is a great suggestion. We will look into evaluating over stochastic environments. As our approach relies on probabilistic improvement guarantees, as the problem becomes noisier (stochastic) it will simply require additional evaluations in order to ensure high probability improvement.

---

### Official Review · AnonReviewer3 · 2019-10-23
**Official Blind Review #3**

**Rating:** 3

**Review:**


[Summary]
This paper proposes an approach called conservative policy gradients to stabilize the training of deep policy gradient methods. At fixed intervals, the current policy and a separate target policy are evaluated with a number of rollouts. The target policy is then updated to match the current policy only if the current policy is better than the target policy. Experiments show that the proposed method, when applied to TD3, reduces the variance in performance through the training.

[Decision]
I am not convinced that the proposed method is sound and indeed useful and I vote for rejecting this paper. Experiments show stable performance. However, this stability comes at the cost of extra computation and interaction with the environment (to evaluate the policies). Claims about the method's stability guarantees and overall performance are not supported by theory and experiments. The submitted paper also needs major improvements in presentation.

[Explanation]
While Proposition 1 provides insights on how the policy evolves, it is too limited to serve as a guarantee. First, performance does not improve or degrade by a constant number. Second, the time it takes for the policy to improve is not captured by the theory. In reality, this time can depend on the hyperparameters or the policy's current performance and might even be unbounded.

I do not understand why a characterization of the performance in the limit of time in Proposition 1 is called a stability guarantee while in the rest of the paper stability refers to consistent improvements in the interim performance. Does stability in this proposition mean that the performance will reach a stationary distribution with bounded support? This property is merely a result of the assumptions that the performance evolves by a constant number at bounded times and that it does not exceed [v_min,v_max].

The theory studies the stationary distribution of the target policy's performance but the algorithm uses the online policy to interact with the environment. In Algorithm 5, line 8 (Section D in the Appendix) the target policy is only used for bootstrapping. How can a stable target policy result in more stable performance if it is not used to take actions?

The paper claims that the proposed method results in improvements in stability and overall performance. In Figure 3, the proposed method is more stable than the baseline but the overall performance is not better.

The proposed method requires more computation and interaction with the environment than the baseline. The experiments do not seem to compare these two methods with the same number of samples or with the same amount of computation. Perhaps the extra computation and samples are better spent on training TD3 for a longer time.

I find Section 2 hard to follow. This section describes Value Iteration, Policy Iteration, DQN, and DDPG in detail (with pseudocode) along with their convergence rates. The message that deep RL algorithms generally lack theoretical guarantees can be conveyed by just describing the linear and deep variants of one method. In fact, the algorithm whose stability is analyzed in the next sections, TD3, is not described in Section 2 or anywhere else in the paper.

Later in Section 3, DDPG and DQN are described as off-policy Deep RL variants of Value Iteration and Policy Iteration. DQN and DDPG are actually built on Q-learning and Deterministic Policy Gradient (DPG).


[Minor comments]
In the learning curves in Figure 1, what is the measure of performance and how is it estimated? A description of the plotted measure is necessary to show that the drops in the estimated performance are indeed due to policy degradation rather than poor estimation.

--------------------
After rebuttal: I have read the authors' response, the other reviews, and the revision. The revised version has improved presentation, but the proposed method is still introduced as a method with stability guarantees while the proposition in the paper cannot serve as a stability guarantee, and can only provide intuition on the asymptotic performance.


**Experience Assessment:**

I have published one or two papers in this area.

**Review Assessment: Checking Correctness Of Derivations And Theory:**

I carefully checked the derivations and theory.

**Review Assessment: Checking Correctness Of Experiments:**

I carefully checked the experiments.

**Review Assessment: Thoroughness In Paper Reading:**

I read the paper at least twice and used my best judgement in assessing the paper.

---

> ### Author Response · Authors · 2019-11-07
> **Official Response**
>
> Thank you for your thorough review. We will address your points below:
>
> - Proposition: Indeed as you have mentioned, the Proposition analyzes the asymptotic performance. The goal is to provide intuition - by performing these evaluations and ensuring that there is a high enough probability of improvement, the policy is more likely to continue improving and achieve a stable solution. This is in contrast to other deep rl approaches which are highly unstable.
>
> - Stability: The target policy is used both as a stable measure for training the critic and at line 13,14 when the policies are evaluated and swapped if needed. Line 8 is also the approach in TD3/DDPG, and as we show in the preliminaries it is inspired by policy iteration techniques.
>
> - Results: While our approach did not improve the mean performance across all domains, this is the case in most of them. Hopper, Inverted Pendulum, Inverted Double Pendulum and Walker saw improvement both in mean performance and in variance reduction. HalfCheetah is slightly worse. And indeed in Ant we saw a reduction in performance.
>
> - Computation: In Figure 3, the X axis denotes the number of gradient steps. We will also add simulations for TD3 with a longer training regime, to demonstrate that even with a similar number of samples, our approach improves the overall performance.
>
> - Section 2: This is also a remark we received from the other reviewers - one which we intend to fix. We will rewrite this section in order to make it clearer. As you noted, DQN and DDPG are derived from Q-learning and DPG, but their origins go further back to VI/PI.
>
> - Figure 1: The performance in all our graphs was estimated over 100 episodes. Meaning, each 1,000 steps we evaluate the policy for 100 episodes and plot the average performance (per seed). As previous works performed either a single evaluation or 10, we see it as highly unlikely that these drops occur due to poor estimation.

---

### Official Review · AnonReviewer1 · 2019-10-24
**Official Blind Review #1**

**Rating:** 1

**Review:**

This paper proposes a simple method for stabilizing the off-policy deep reinforcement learning algorithm, which updates the target network only when the online network performs better than the target network in order to ensure the stability guarantees. More specifically, at every T time steps, they execute both the online network and the target network so as to evaluate the performance of each policy. Then, they update the target network only when the online network outperforms the target network with high probability. The experimental results show that the proposed Conservative-TD3 (C-TD3) is less prone to performance degradation during training.

While the stabilizing off-policy reinforcement learning algorithms would be a significant problem, I have some concerns regarding the presentation and the limitation of the proposed method.
- In p2, 'Value Iteration and Policy Iteration are algorithms for solving tabular RL tasks with convergence guarantee': Basically, exact value iteration and policy iteration are MDP 'planning' algorithms, NOT reinforcement learning algorithms. VI and PI assume 'known' transition and reward dynamics thus there is no need to learn anything, while RL basically assumes the 'unknown' environment thus the agent should learn by doing.
- Algorithm 1, 2, 3 and 4 are not directly related to the proposed method, thus they can be omitted from the paper. Instead, it would be great to devote more space to the proposed method such as a more detailed theoretical analysis or the pseudo-code of the proposed algorithm.
- It seems that the performance of the online and target networks is evaluated by Monte-Carlo return which is obtained by executing each policy in the real environment. This requires additional direct interaction with the environment, which can severely hurt the sample complexity of the algorithm. In Figure 4, does the x-axis of C-TD3 reflect these additional samples for evaluating the performance of two policies?
- The abstract says 'our proposed method reduces the variance of the process and improves the overall performance'. This claim is too strong. If we see Figure 4, in Walker2d, the mean performance of TD3 reaches 4000 at 400k steps, while the performance of C-TD3 is even less than 3000. Similarly in HalfCheetah and Ant, the asymptotic performance of TD3 is higher than C-TD3. It would be great to show the learning curves of TD3 and C-TD3 overlapped.
- In the experiments, 'discarding failure seeds' cannot be a proper treatment. Instead of discarding the bad results, the reliable algorithm had to be proposed.

**Experience Assessment:**

I have read many papers in this area.

**Review Assessment: Checking Correctness Of Derivations And Theory:**

I assessed the sensibility of the derivations and theory.

**Review Assessment: Checking Correctness Of Experiments:**

I assessed the sensibility of the experiments.

**Review Assessment: Thoroughness In Paper Reading:**

I read the paper at least twice and used my best judgement in assessing the paper.

---

> ### Author Response · Authors · 2019-11-07
> **Official response**
>
> Thank you for your review.
>
> We understand that the main issue is in how the method was presented. As there is a tight connection between VI/PI and Q-learning/Policy Gradient methods, we thought that this explanation would be easier to comprehend and the motivation would be clearer (a similar approach was made in the TRPO work, which built upon Conservative Policy Iteration).
>
> However, we understand that this caused more harm than good - thus we would like to state that during the rebuttal period we will be working on rewriting the motivational sections. We will post an additional update once this is done.
>
> Regarding the other remarks.
> - Theory: The goal of the theoretical analysis was to provide intuition. Intuitively, as the probability of improvement increases, then the stable distribution will place more probability mass at being at a higher value.
> - Pseudocode: As the approach is not complex, we believe that the flow-chart is easier to comprehend, thus we provided the pseudocode in the appendix. However, if we do end up removing Alg. 1-4 in the rewrite, we will follow this advice and place it in the main portion of the paper.
> - Evaluation: Indeed the evaluation is performed using Monte Carlo sampling. Just as you have stated this leads to additional environment interactions, we will make sure this is clearer. The X axis in this case corresponds to the number of gradient steps. In our opinion, these are intruiging results which give motivation for further research - for instance find ways to perform off-policy policy evaluation in order to determine which policy is better, i.e., removing the additional environment interactions.
> - Claim: We will tune down the claim. While the results did reduce variance across all domains and the asymptotic performance across almost all domains was matched/improved, indeed this is not in all domains and the abstract should reflect these results.
> - Failure seeds: We also feel somewhat uncomfortable with this approach, but we believe it is better to be up-front rather than hiding it. These failure seeds occurred both in the C-TD3 and TD3, at a similar frequency, where TD3 is the original code provided by the papers authors; as such, since both algorithms received the same "care" and since this is not the focus of our work, we believe that this approach yields a fair comparison.

---

### Decision · Program_Chairs · 2019-12-19

**Decision:**

Reject

**Comment:**

This paper proposes a new algorithmic approach to reduced variance in off-policy, policy gradient updates.

Multiple reviewers were concerned with both the soundness of the proposed approach, and the cost of using rollouts. In particular, the interaction between the target policy and the behavior policy, and how they are swapped was unclear, where the algorithms in the paper do not match the code provided.

The results show apparent reduction in variance across runs compared with TD3: clear improvements in two domains, minor improvements , and/or an increase in variance in others. In some domains there was decrease in mean performance. The reviewers wanted comparisons with other baseline methods (even in terms of variance across runs).

It is difficult to evaluate the results in this paper, as the performance is averaged over only 5 runs, and runs which result in "failure" are discarded from analysis. The authors explain this was done in the original TD3 code, and one can sympathise in following common practices in the literature. However, the consensus of the reviewers and the AC was that this choice was not well defended, obscures a key difficulty of the learning problem, and makes algorithms look considerably stronger then they actually are. This is particularly confounding in a paper about improving the robustness of learning algorithms. This is not acceptable empirical practice and we strongly encourage the authors to discontinue this.

The reviewers gave nice suggestions including changing the pitch of the paper, and including results in noisy tasks. To reduce the burden of doing more scientific experiments, we suggest the authors start with small or even designed problems to carefully study robustness of learning and the potential improvements due to their algorithm. After this is done in a statistically significant way, it would be natural to move to more demonstration style scaled up results.